# Online Mathematics Teacherpreneurs Developers on Teachers Pay Teachers: Who Are They and Why Are They Popular?

**Amanda G. Sawyer** [1,*] **, Lara K. Dick** [2] **and Pierre Sutherland** [3]

[1] Department of Middle, Secondary and Mathematics Education, James Madison University, Harrisonburg, VA 22807, USA

[2] Department of Mathematics, College of Arts & Sciences, Bucknell University, Lewisburg, PA 17837, USA; lara.dick@bucknell.edu

[3] Department of Mathematics, Clevedon School, Clevedon BS21 6AH, UK; psutherland@clevedonschool.org.uk

* Correspondence: sawyerag@jmu.edu

**Abstract:** Online teacherpreneurs are teachers who use social media platforms to create, sell, and distribute educational resources to others. For many teachers, they have become the new curriculum developers in our virtually intensive world. Curriculum development in mathematics education has a large impact on how students understand concepts, but little is known about these online mathematics teacherpreneurs influences on the curriculum. Therefore, as part of a larger study investigating the top 500 free elementary mathematics educational resource developers on TeachersPayTeachers.com (TpT), we surveyed the teacherpreneurs who created these top resources. Using the constant comparative method, we analyzed 58 responses to learn more about the online mathematics TpT teacherpreneurs: who they are, how they believe they became popular, and what they believe their teacher customers want when searching for resources. We found these teacherpreneurs identified themselves as teachers with typically over 10 years of experience creating classroom resources. Many attributed their popularity to advertising their resources via social media and having a large number of products available for teachers. They identified beliefs that teachers want easy to use, free, quality materials that are visually appealing. Implications, including findings that indicate a misalignment between what teachers say they want and what the teacherpreneurs believe teachers want, are discussed.

**Keywords:** social media; teacherpreneurs; Teachers Pay Teachers; curriculum developers

## 1. Social Media and Mathematics

The influence of social media on teachers has moved beyond their personal lives to influencing their professional lives, partially because of the growing trend of individuals promoting their self-made educational products online. The promotion of these types of educational resources has grown exponentially on social media in recent years [1,2]. Teachers learn about teaching resources from professional organizations and from the marketing of publishing companies, but teachers are also increasingly exposed to advertisements for curricular resources via social media sites like blogs, Instagram, Twitter, and Facebook. Current and former teachers who market their own resources through social media are referred to as online teacherpreneurs [3]. This exchange can blur the line between personal and professional interactions where the online teacherpreneurs creating and promoting self-made curricular resources are also the teacherpreneurs' colleagues and friends.

For this study, we define social media as websites that support communication through member-created profiles [4]. Thus, social media includes sites such as Facebook and Instagram, but also

professional-based sharing sites such as TeachersPayTeachers.com (TpT) and the newly launched Amazon Educational Insight. These social media sites allow teachers to create a profile, share educational resources, and communicate with each other. In 2018, through surveying 601 United States teachers, Shapiro et al. found teachers reported TpT to be the most popular social media site, with 89% of elementary mathematics teachers surveyed identifying using TpT in their classrooms [1]. Because of its popularity and the ways teacherpreneurs can utilize the TpT platform to promote their educational materials as a money-making endeavor [2], for this paper, we are specifically interested in online teacherpreneurs who share resources on TpT.

*Teachers Pay Teachers*

TpT was founded in 2006 by a New York public school teacher, Paul Edelman [5]. On the TpT website, anyone can post self-created curriculum materials including manipulatives, worksheets, games, and posters for free or for a self-designated fee. Individuals who create mathematical resources on TpT, from a single worksheet to a set of task cards, can be called online mathematics teacherpreneurs. TpT has grown exponentially since its creation and was awarded the 2019 most innovative company by the business magazine Fast Company [5]. In the spring of 2019, TpT reported more than 5 million teacher users, over 3 million available resources with over 1 billion resources downloaded from their website. In addition, TpT claims "more than two out of three teachers in the U.S. have used a resource on Teachers Pay Teachers" [5]. Because many teachers need to supplement their income, some have chosen to do so by becoming online teacherpreneurs [3,6]. Shelton and Archambault found monthly earnings from TpT teacherpreneurs ranged from $5 to $68,000, and Reinstein found that teachers' promotions of their educational resources amassed over 100,000 followers and resulted in over $200,000 of profit each year for some creators [3,5].

The field of education has just begun to research the online teacherpreneurs who contribute to social media sites such as TpT. Shelton and Archambault found that the general population of teacherpreneurs on TpT were predominantly highly educated, white, female, experienced teachers [3]. Their study was not subject specific, but this study seeks to find out more about elementary mathematics online teacherpreneurs. Mathematics education is a field where the curriculum influences the understanding of the concepts [7]. The mathematical task itself can determine the level of cognitive demand a student employs [7]. Through previous research looking at elementary math resources on TpT, we have learned that the curriculum materials on TpT and other social media websites typically have low cognitive demands [2,3] indicating a need to know more about these online teacherpreneurs.

## 2. Literature Review

Past research into curriculum development has centered around three major elements: creation, adaptation, and implementation of educational resources in classrooms [8–10]. However, this has changed since the advent of online teacherpreneurship because the individuals who are creating, adapting, and implementing the resources are now accomplishing all three elements themselves [3]. In this review of literature, we discuss what is known about online teacherpreneurs. In addition, we share literature on curriculum development more generally and finally discuss online mathematics curriculum development more specifically.

*2.1. Online Teacherpreneurs*

In years past when referencing curriculum developers, the education community referred to big companies like Pearson, Houghton Mifflin, and other sellers of published textbooks, but in the advent of sharing marketplaces on social media websites, there is a new group of individuals referred to as online teacherpreneurs who sell their self-made curriculum materials online. Shelton and Archambault defined online teacherpreneurs as current or former teachers that market their own resources [3]. It must be noted that online teacherpreneurs are not the same as how Berry defines teacherpreneurs [11]. Berry views these individuals as teacher leaders and "classroom experts who

teach students regularly but also have time, space, and reward to spread their ideas and practices to colleagues as well as administrators, policy makers, parents, and community leaders" [11] (p. 146). In this paper, we investigated curriculum developers of self-posted and promoted online materials, online teacherpreneurs, rather than teacher leaders as described by Berry [11]. The online mathematics teacherpreneurs could be teacher leaders in their field, but we are only viewing one aspect of their work which includes their creation and posting online educational resources on TpT, thus we cannot know their teacher leadership status.

As noted, TpT's marketplace for teachers to sell their products was created in 2006, and further marketing of these resources was increased with the launch of the social media site Pinterest in 2010. With Pinterest, teachers could pin TpT resources to boards and advertise products [5,12]. The TpT marketplace is a hugely growing phenomenon with much earning potential for online teacherpreneurs [13]. For this investigation, we define the online mathematics teacherpreneurs under study similar to online teacherpreneurs, but we admittedly do not know if they are current or former teachers. Even though the TpT's name identifies the teacherpreneurs as teachers, anyone can post their resources, thus we believe more research is needed to determine the background of these teacherpreneurs.

To begin to delve into the landscape of TpT teacherpreneurs, Shelton and Archambault surveyed TpT sellers of all K-12 curriculum materials to determine characteristics of online teacherpreneurs and identify the impact of their resource creation on their teaching practice [3]. They found that the teacherpreneur population was predominantly white females with about 40% retired teachers. These surveyed online teacherpreneurs took great pride in their work in constructing curriculum materials and separated their school work and business as an educational resource developer. The teacherpreneurs reported believing that their teaching practice was improved from their constructions of classroom materials [3]. In this investigation, we continue to contribute to this landscape by focusing on elementary mathematics online teacherpreneurs, both in studying their demographics and in determining what they find is important when creating and promoting their resources.

*2.2. Curriculum Development and Classroom Implementation*

Individuals within and out of the teaching profession can create educational curriculum and share it on social media sites. Little is known about non-teacher curriculum developers. However, physical curriculum development by classroom teachers has been researched; research shows the process teachers go through when developing curricula to be a complex process that reflects teachers' perceptions of how concepts should be taught [7,14,15]. Once developed, different factors such as knowledge, local contexts, personal identities, and beliefs shape how teachers read, interpret, and implement curricula [16]. According to Davis et al., teachers should be involved in designing curriculum because they can provide insights into the specific needs of their students [16]. Despite these findings, Ding and Carlson found that many teachers do not alter curriculum resources developed by others because they are concerned that their adjustments will not align with district or state standards [15]. The study also found that in terms of lesson planning, teachers fall on different parts of the spectrum. Some strictly follow the curriculum materials given to them while others adapt curriculum materials to their needs.

Many academic publications as early as Bidwell suggest that teachers should be more involved in the construction of physical resources for their classrooms [17]. Carson argued it was actually the teachers right to become their own educational decision-maker [8]. Shawer, Gilmore, and Banks-Joseph continued this understanding by saying that teachers are curriculum developers because of the way they implement the materials in their classrooms [12]. They defined teachers as curriculum transmitters, curriculum developers, or curriculum makers depending upon how much they change the resources. Shawer argued that curriculum transmitters do not change the materials, curriculum developers adapt some materials, and curriculum makers adapt all their materials for their specific classrooms [9]. All three are types of teacher users who download materials created by online teacherpreneurs and posted on social media educational sites such as TpT. Sites like TpT offer a platform where the individuals implementing the materials take part in physically making the resources. This goes along

with the idea set by Carson and Bidwell saying that teachers should be involved in this process, but takes it a step further by making the online teacherpreneur the only person in this process [8,17]. It is important to note that online teacherpreneurs who develop curricular materials are not necessarily developing an entire curricula sequence nor a cohesive set of materials; a TpT teacherpreneur is anyone who creates an educational resource as small as a single worksheet. There is a dearth of research considering how these online teacherpreneurs think about curriculum and resource development, including content specific develop of educational materials; this paper seeks to fill this gap.

*2.3. Teachers Use of Online Resources*

Teachers are using online resources regularly, and the websites they obtain these resources from are changing over time [1,18]. In 2014, the Bill and Melinda Gates Foundation found 91% of teachers use online resources to help plan their lessons, and the most frequently used sites were: Scholastic.com (80%), YouTube (72%), Pinterest (69%), Discovery (64%), and PBS.org (61%) [18]. Researchers in 2016 found that 87% of elementary teachers use Pinterest followed by Google [19]. In 2019, TpT was found to be the most frequently used website (89%) to gain online resources [1].

Teachers used the resources in many ways in the classrooms, but they had many issues with the social media platforms [20,21]. Pollock and Dean found that supplemental materials were popular online, but the popularity of the resource did not reflect if it was a quality resource or if it aligned with standards [20]. Even though teachers find the benefit of using these social media resources in their classrooms, they have difficulties with understanding confidentiality of the resource and were sometimes overwhelmed by the nature of social media tools [21]. Carpenter and Harvey also found teachers had interpersonal difficulty. Specifically, teachers had issues with self-promotion and impersonal content found on social media, and researchers found that the educational community lacked policies and regulations to support the positive aspects and limit the challenges brought about by the social media tools [21].

Many teachers used the social media tools to help identify who they were as an educator [22]. Pittard explained the ever-changing social media marketplace influences how teachers see themselves. Many teachers were trying to keep up with others to be as "good" as what they saw online. Pittard stated teachers were, "revis[ing] and reimagin[ing] how to be successful in their work and lives" [22] (p. 44). This perpetual making-over of what counts as good teaching influences how other teachers think and act, including the types of resources created by online teacherpreneurs.

*2.4. Overarching Investigation on Mathematic Teachers Use of Online Resources*

This investigation into the top online elementary mathematics teacherpreneurs on TpT is a continuation of a larger research agenda constructed by our research group. Research has previously been conducted into general teachers' use of resources online, but we were curious as mathematics teacher educators how mathematics teachers used online resources [18,19]. Thus in 2018, we surveyed 601 elementary mathematics teachers across the United States to investigate their use of online resources [1]. Teachers reported searching for online mathematics resources weekly, regardless of years of teaching experience, and 89% of these elementary mathematics teachers identified using TpT and 74% of elementary mathematics teachers identified using Pinterest [1]. Therefore, we found that mathematics teachers used resources similarly to general teachers [1]. As part of this survey, we also asked the elementary teachers what they looked for in online resources, and we found that elementary teachers believed that alignment to standards, perceived student engagement, and difficulty level were the top three most important elements; the lowest ranking items were the price of the product, visual appeal, and user rating (in descending order) [1]. Schroeder, Curcio, and Lundgren found similar results; their survey of K-12 teachers reported searching for resources based on connections to standards and their view of how resources would best serve their students' needs [23].

Since mathematics teachers were using online resources from websites like TpT and Pinterest most often, our research group investigated the quality of the mathematics materials found on these sites.

Therefore, on June 6, 2018, we downloaded the top 500 free elementary mathematics resources from TpT and Pinterest to determine what types of resources were available, what their level of cognitive demand was, and if there was a correlation between the level of cognitive demand and types of images found on the resources [24]. We found that on both TpT and Pinterest that less than one percent of the resources could be considered the highest level of cognitive demand, and over half of the resources on both TpT and Pinterest were low-level tasks (61% and 63% respectively) [24]. We also found a correlation between "cute," non-functional images and lower-level cognitive demand tasks on both websites [24]. Other researchers like Hertel and Wessman-Enzinger found similar results in terms of lower quality resources on these websites [25].

As mathematics educators, we want to influence the field by increasing the number of high-quality resources found on these websites, which we believe we must do by reaching out to the online teacherpreneurs who are creating and sharing these resources. To do this, we must learn more about who they are, how they became popular, and what they believe teachers are looking for when searching for resources on social media educational websites. Therefore, our next step was to investigate the online mathematics teacherpreneurs who created the top 500 free resources that we collected in 2018. Since most of the pins on Pinterest came from TpT and TpT was the most often reported platform, we decided only to look at the 500 top resources we collected from TpT [1,26].

### 2.5. Research Questions

Many teachers are dependent on online resources because of the push for virtual learning, thus these marketplaces are here to stay and we anticipate will only expand with time. Therefore, it is important to identify who these online elementary mathematical teacherpreneurs are, how they become popular, and what they believe about the teachers who search for and implement their resources. To begin to answer these questions, we surveyed the online teacherpreneurs who created top 500 free elementary mathematics resources on TpT, and from the responses received, we answered the following research questions.

1.　How do online mathematics teacherpreneurs on TpT identify themselves?
2.　What do online mathematics teacherpreneurs on TpT believe has led to their popularity?
3.　What do online mathematics teacherpreneurs on TpT believe their customers/teachers desire in elementary mathematics resources?

From this data, we were able to learn more about who these online mathematics teacherpreneurs are and what they believe is important for creating mathematical resources to share with other teachers. The mathematics curriculum that students experience influences their understanding of mathematics, therefore the field of mathematics education can benefit from knowing more about the teacherpreneurs who create these curriculum materials.

### 3. Methods

To investigate this phenomenon, we used survey methodology in which participants were asked to answer predefined criteria about their teaching experience, self-identified career, and years of producing mathematics curriculum. The survey also offered open-ended questions for participants to describe how they view their TpT popularity and their beliefs about what their teacher customers desire. The full survey is available in Appendix A.

### 3.1. Sampling Technique

As mentioned, we previously investigated the top 500 free elementary mathematics resources downloaded from TpT on June 6, 2018 [24]. We defined top by using the following filters provided by TpT: top-rated, free, grades: K–5, and math. In 2018, top rated meant a resource had four out of four stars for overall quality, accuracy, practicality, thoroughness, creativity, and clarity. It must be noted that in 2019 the scale went up to five stars, but when we collected these 500 resources the

four stars were the top rating. We chose to use ratings because research shows that teachers use ratings to select resources and we chose to select free resources because we wanted to observe popular materials available to all teachers despite the price [27].

To learn about the online teacherpreneurs who developed these top free resources, we gathered the screen names from the teacherpreneurs associated with the top 500 resources, and contacted them through TpT, asking them to complete the survey. We found 321 separate TpT teacherpreneur users created the top 500 resources, and thus contacted the 321 teacherpreneurs through their TpT profile. TpT does not provide a way to privately message each person; however, it does allow a TpT user to select the tab, "Ask a Question." Questions are typed into a box and submitted, which automatically emails the creator, and posts the question on the TpT "question and answer section" of the profile page. In this question box, we wrote a letter explaining our research, shared our Institutional Review Boards information related to the project and asked these top TpT elementary mathematics teacherpreneurs to complete the survey. On June 26, 2019, all 321 teacherpreneurs were contacted in this manner, but after posting on their profile pages, we were contacted by TpT administrators and told that we could not continue to contact their users because it was viewed as solicitation. Because of this difficulty, we were not able to recontact the teacherpreneurs to ask for more participation in our survey. From this one attempt at reaching the online teacherpreneurs, we received 58 survey responses. All fifty-eight shared at least some basic demographic information. However, only thirty-six of the 58 participants responded to the open-ended question related to how they became popular/successful on TpT, and thirty-nine of the 58 responded to the question of what they believe teachers are looking for when selecting resources.

*3.2. Data Analysis*

To answer the question of who these online teacherpreneurs are, we used descriptive statistics to provide all frequencies and percentages related to the total participation. We collected data from the survey and displayed the frequency and percentages of the responses. To answer the question about what the online teacherpreneurs believed contributed to their popularity, two researchers used the constant comparative method to analyze their open-ended responses [28]. From this process, eight categories were identified that captured the responses. These categories were social media, quality, niche, early to TpT, large number of products, offer free things, relatable, and don't know. Participant responses could be coded with multiple categories. The same two researchers also used the constant comparative method to analyze the qualitative data describing how the online teacherpreneurs viewed what their customers desire in educational materials. Ten different categories from the responses were identified: easy-to-use, visual appeal, price, rigor, quality, alignment with standards, engagement, rating, technology, and differentiation. Again, participant responses could be coded with multiple categories.

After identifying the categories for all responses, the researcher developed a detailed codebook using a process as described by DeCuir-Gunby, Marshall, and McCulloch, and shared it with the third researcher (see Appendices B and C for the full codebook) [29]. To assure validity of the identified categories and their coding, 10 creator survey responses were chosen at random and the third researcher applied the codebook to categorize the responses. Two new codes were created from this calibration. For the research question regarding why the participants were popular, we added the category "fair price", and for the research question looking into what the teachers want in resources, we added the category "fits a need". After coding, eighty-eight percent agreement was reached and therefore it was determined that the codebook was reliable. The original researcher then re-coded all responses with the newly identified codes. Once coding was complete, frequencies were determined for the categories and exemplars were identified for the discussion of the results.

## 4. Results

### 4.1. RQ1: Who are the Online Mathematics Teacherpreneurs on TpT?

All 58 participants responded to the research question of, "Are you a current teacher?" They could respond with either, (a) Yes, I currently teach elementary school, (b) Yes, I currently teach something other than elementary school, (c) No, I am a former teacher, or (d) Other: ______. The majority of the participants identified themselves as current or former teachers, as seen in Figure 1. Twenty-seven (46.6%) individuals responded that they are currently teaching elementary school, and three (5.2%) individuals explained they teach something other than elementary school. Nineteen (32.8%) participants stated they were former teachers with eight (13.8%) identified an occupation other than teaching. TpT identified itself as a website where educators are creating resources for other teachers, but we found that only 84.6% identified themselves as teachers. The non-current teachers provided occupations such as "retired military", "homeschool my two children", or "I am a school psychologist" while many did not provide details and instead chose "other". Only one of the 58 participants chose not to answer the question at all.

**Figure 1.** Online Teacherpreneur Teaching Status.

Fifty-three of the 58 participants responded to the question of "If applicable, how many years of teaching experience do you have" with responses of 0–2, 3–5, 6–10, 11–15, 16–20, 21–25, 26–30, 31–35, and 36+. While not all the participants were current teachers, all except 5 reported some teaching experience with many teachers reporting teaching for many years (Figure 2). We found 39 (67.2%) of the participants had more than 10 years of teaching experience and 14 (24.1%) participants had less than 10 years of experience, and the largest population (12 participants) had between 11–15 years of experience. Perhaps surprisingly, we had more teachers with more than 26 years of teaching experience (9, 15.5%) than less than 5 years of experience (5, 8.6%).

In addition to years' teaching experience, we also asked the participants, "How many years have you been producing teaching resources?" and "How many years have you been producing teaching resources for TpT?" (Table 1). All 58 participants responded to these questions with the largest response for producing resources was 15 years for 27.6% of the participants. Twenty-four (41.4%) participants indicated that they began creating resources before producing resources for TpT. When asked if they created curriculum materials for any other website, 42 (72.4%) participants said they only created materials for TpT, but 11 (19%) said they had produced for other websites. Two participants identified

that they were affiliated with a publishing company, but they did not identify which companies they worked for despite being asked to provide that information.

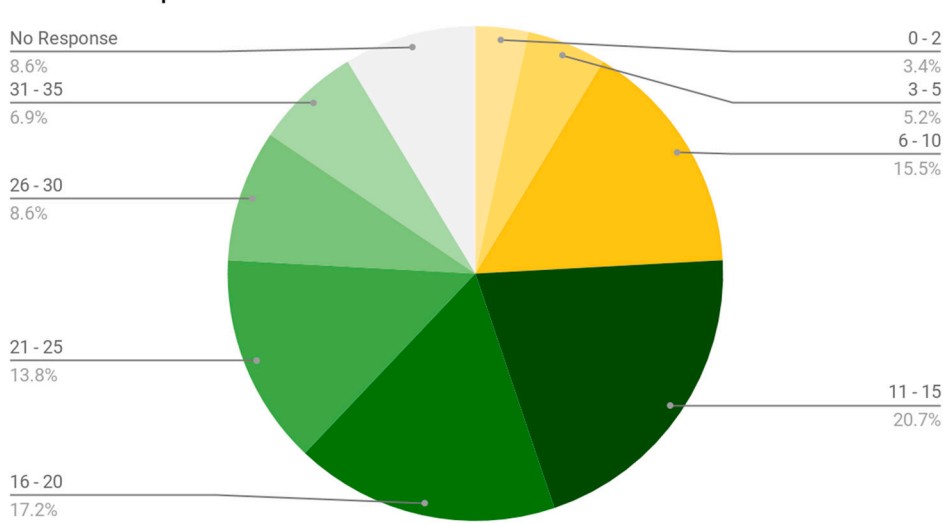

**Figure 2.** Online Teacherpreneur Years of Teaching Experience.

**Table 1.** How TeachersPayTeachers (TpT) Teacherpreneurs Explain How Many Years They Produced Resources.

| Years | How Many Years Have You Been Producing Teaching Resources? | | How Many Years Have You Been Producing Teaching Resources for TpT? | |
|---|---|---|---|---|
| | Frequency | Percent | Frequency | Percent |
| 2 | 2 | 3.4 | 2 | 3.4 |
| 4 | 2 | 3.4 | 2 | 3.4 |
| 5 | 3 | 5.2 | 8 | 13.8 |
| 6 | 10 | 17.2 | 13 | 22.4 |
| 7 | 2 | 3.4 | 13 | 22.4 |
| 8 | 5 | 8.6 | 9 | 15.5 |
| 9 | 2 | 3.4 | 4 | 6.9 |
| 10 | 4 | 6.9 | 3 | 5.2 |
| 11 | 1 | 1.7 | 0 | 0 |
| 12 | 4 | 6.9 | 0 | 0 |
| 14 | 3 | 5.2 | 0 | 0 |
| 15 | 16 | 27.6 | 0 | 0 |
| No Response | 4 | 6.9 | 4 | 6.9 |
| Total | 58 | 100 | 58 | 100 |

*4.2. RQ2: What do They Believe Has Led Them to Be One of the Top 500 Mathematical Resource Teacherpreneurs on TpT?*

To investigate the popularity of these online teacherpreneurs, we asked the following question, "Please explain how your popularity and number of followers have grown over time. What do you believe made this occur?" Of the 58 participants, 36 answered this question and provided a variety of reasons (Table 2). The top two responses were offering quality resources and promotion on social media. Other responses included offering a niche in the market, beginning early on TpT, offering many products, offering free products, and being relatable to their customers. Three (8.3%) of the participants indicated they did not know the reason for their popularity.

**Table 2.** How TpT Teacherpreneurs Explained Their Popularity.

| Codes | Frequency | Percentage |
|---|---|---|
| Quality | 17 | 47.2 |
| Social Media | 16 | 44.4 |
| Offer Free Stuff | 7 | 19.4 |
| Early to TpT | 5 | 13.9 |
| Large Products | 6 | 16.7 |
| Niche | 4 | 11.1 |
| Don't Know | 3 | 8.3 |
| Fair Price | 3 | 8.3 |
| Relatable | 2 | 5.6 |

### 4.2.1. Quality

The largest population (47.2%, 17 responses of the 36) of participants identified offering quality materials as an explanation of their popularity on TpT. They took pride in the time they took creating and "revamping" their items for their customers. For example, one participant explained:

> I work very hard to create high quality resources and make them available to teachers. I wouldn't say that I'm "popular", and I don't think I have any "followers" at all… I'm not an Instagram influencer. I'm a teacher with a [Master's] degree…and 14 years of classroom experience working with kids from all kinds of backgrounds and abilities…"Popularity" and "followers" don't really help kids learn that well, in my experience. (Participant 31)

By providing the best products "made by teachers", they felt that they offered something unique that was not available in traditional curriculum materials.

### 4.2.2. Social Media Promoted

In contrast to the response above, the next largest majority of the participants who responded (44.4%, 16 responses of the 36) discussed using social media to increase their popularity. For example, participant 23 responded, "My growth has [been] slow and steady. It's directly related to my production and social media marketing." The participants identified specific elements in their social media promotions that helped them become popular. This included three different categories: using TPT's features to advertise, using their own website to advertise, or promoting their curriculum on other social media websites.

Four participants discussed using TpT's advertising, promotions, and features to help increase their product's popularity. TpT teacherpreneurs have the opportunity to apply to become featured producers which allows them to become more known; this feature was discussed by participant 19 who stated, "I also find applying for TpT promotion and exposure has greatly helped…there's so many sellers and even more resources that it's hard for teachers to find you even if you have incredible products." By applying for promotions, TpT teacherpreneurs can be chosen to have their resources featured on the website to promote their products.. The importance of this feature was also discussed by participant 24 who stated, "I submitted some of my free resources to be TPT features and have successfully been featured a couple of times over the years, which has been helpful in acquiring followers." In addition to promotion, some of the participants indicated a reason for their popularity was learning how to use specific keywords and phrases that users would be looking for to maximize their products appearing after a TpT search. For example, one participant stated, "Using product names that include the skill or concept name, believe, has helped teachers find my products when they search on TPT" (Participant 27). All of these examples highlight the teacherpreneurs' beliefs that their popularity can be explained, at least partially, by their strategic use of TpT functions and features.

Outside of TpT, some participants indicated promoting themselves on their own blogging websites as a means to increase their popularity. Five participants (13.9%) discussed how their blogs were a big

element of their social media presence and how from their blogging websites they advertise contests and promotions that link to their TpT store. For example, participant 2 stated, "Having a blog and being active by posting and sharing free items as well as paid items got my attention. Also, joining link parties and having contests or giveaways to gain followers [explains my popularity]."

In addition to their own websites and strategic use of TpT, many participants reported advertising on other social media platforms such as Pinterest, Facebook, and Instagram as a way to increase their popularity on TpT. These participants mentioned things such as using group boards and Facebook groups to feature their self-made curriculum materials on these websites. For example, participant 21 stated, "Years of blogging, advertising on Facebook, posting on Instagram, working the trends, co-authoring products with other teachers, hard-work, as well as dedication all have contributed to my success." Pinterest as advertising for the TpT was a common social-media theme across this group with six participants (16.7%) naming the website to help promote their resources. For example, participant 15 stated, "I think the initial use of Pinterest to pin my resources to multiple boards and groups helped and I'm seeing the effects of that years later." While other participants just discussed being, "very active on Pinterest" or other sites, not all teachers believed their presence on social media necessarily increased their sales. One participant explained:

> Social media, such as FB and Pinterest helped a bit to get my brand name out there and advertise a bit . . . It's better if you grow your followers organically. I never do social media anymore for my store. It may result in a bit less traffic to my store, but the time required to maintain a social media presence is not worth the very few sales it may generate. Ninety-eight percent of my sales come organically from within TPT, not from outside the site. (Participant 25)

This participant indicated a belief that while social media can increase traffic on TpT, they did not believe the effort to advertise on social media led to higher teacherpreneur income on TpT.

### 4.2.3. Early to TpT

Five (13.9% of the 36 responses) of the participants discussed being some of the initial teacherpreneurs to join the TpT social media site. For example, one person explained, "I got involved with TPT early on and now there are so many other TPT authors that it is hard to get a larger following" (Participant 7). They attributed this "early" arrival to their growing success because they were able to be featured several times by the website itself. Another individual explained:

> When I first started on TpT it was still a very small community of sellers and my followers and sales grew quite fast. I believe I have retained my early followers, and gained new ones over the past years. As I am not very involved in social media and blogging I rely a lot on the marketing that TpT does. My top freebies are often featured in newsletters and emails to customers; I believe that this exposure brings people to my store. (Participant 36)

These individuals all discussed how their years of working with TpT has helped them despite not being as involved in their social media web presence. For example, one participant stated:

> I started when no one knew about TpT. I created resources for my classroom for 20 years, and a friend said I should sell them. I only have 14 products, but 1008 followers. I know I need to make more resources, and I would sell more. But so many people are on Twitter, Facebook, etc., I cannot spread myself so thin. (Participant 21)

By being the first on this social media-based marketplace, they were able to build this following that has continued to grow.

> At the time when I started creating products (2012), I was one of the few that differentiated activities and focused on developmentally appropriate activities such as sensory, fine motor, movement, and game play. These helped me create a niche that has since been saturated by teachers that came after me, but I was one of the first... and that's where most of my followers came from when I look back. (Participant 13)

4.2.4. Large Number of Products and Free Products

Sixteen and seven tenths percent (6 responses of the 36) identified creating a large number of products and 19.4% (7 responses of the 36) believed offering free resources helped their popularity. For example, one individual explained:

> Popularity skyrocketed when I had my first child and was on maternity leave; I was able to focus on creating resources and "stocking up" my store. It also increased as my skills developed and I went through older resources and revamped them (consistency between quality, product images, etc.) to match my most recent resources. (Participant 28)

> Once they had a large number of products, they offered "freebies" to help entice new customers. I offer new free items occasionally which I know my followers appreciate and it helps draw in new customers as well. I keep my prices as fair as I can because I understand what it's like to work in environments that give teachers few resources or materials that use up their budget. (Participant 33)

This showed the importance of free items as well as a large number of products to bring people to their TpT store.

*4.3. RQ3: What do They Believe Their Customers/Teachers Are Looking for in Elementary Mathematics Resources?*

To learn more about what drives the creators' development of classroom resources, we asked the open-ended question, "What do you think teachers look for when they are searching for elementary mathematics materials?" We received 39 responses from the 58 participants as seen in Table 3. Twenty-one (53.8%) participants believed ease of use was most appealing to teacher consumers. These teacherpreneurs wrote responses such as:

> I hope they look for engaging developmentally appropriate activities that align with standards and make learning fun. In reality I am pretty sure that many look for easy (for the teacher to use) activities that keep the students busy. I am sad to say that was my experience with many colleagues before I retired. (Participant 31)

Nine (23.1%) participants believed it was visual appeal that had their consumers purchasing their items. Even though they knew that is not what is best for the classroom, they responded with, "Unfortunately I think teachers overvalue visual appeal on TPT, as well as ease of prepping" (Participant 12). Five (12.8%) participants believed the price was a large factor in what teachers are looking for. For example, one person responded, "Affordability" (Participant 25), while others discussed "I THINK TEACHERS JUST WANT FREEBIES!!!" (Participant 29).

**Table 3.** How Teacherpreneurs Explain What Teachers Are Looking For in Mathematics Resources.

| Codes | Frequency | Percentage |
| --- | --- | --- |
| Easy to use | 21 | 53.8 |
| Alignment with Standards | 16 | 41.0 |
| Fits a Need | 14 | 35.9 |
| Engagement | 12 | 30.8 |
| Visual Appeal | 9 | 23.1 |
| Differentiation | 6 | 15.4 |
| Price | 5 | 12.8 |
| Rigor | 5 | 12.8 |
| Quality | 5 | 12.8 |
| Rating | 2 | 5.1 |
| Technology | 2 | 5.1 |

Five (12.8%) participants believed that rigor and quality were important elements. These participants believed teachers want materials that "challenge students' thinking" (Participant 1).

Sixteen (41%) participants believed alignment to standards were important. This included teachers wanting "common core alignment [materials] in a kid friendly style" (Participant 32). Twelve (30.8%) participants believed level of student engagement is important to teachers who purchase their items. For example, one participant responded: "I hope they are looking for hands-on, engaging, and differentiated activities that allow their students to be successful and independent" (Participant 9).

There were a few qualities that were discussed only by a couple of participants including technology and ratings. For example, Participant 3 stated that teachers wanted, "Digital resources. Resources that use the latest technology". While the participants who mentioned rating typically stated it as a question. Participant 29 responded, "Rating, curriculum, I'm not sure. If I knew, maybe I could sell more".

## 5. Discussion

In this discussion, we answered the research questions in the order they were asked. First, we discussed what we learned about participant demographics explaining what we learned about these online teacherpreneurs. Second, we described how these online teacherpreneurs became popular and what elements they use to increase their popularity. Third, we explained what the online teacherpreneurs believe their teacher consumers want which indicated a discrepancy with what we found in our earlier study regarding what teachers actually stated they want from online resources.

### 5.1. Demographics

Our analysis of the participants showed the population of top online elementary mathematics teacherpreneurs on TpT were very similar to the population of general creators of TpT found by Shelton and Archambault [3]. These individuals were experienced, current, and former teachers who created their own curricular materials. We also found the number of teacherpreneurs that are currently not teaching to be proportionate to the 40% of the retired population in Shelton and Archambault's survey [3]. In addition to these findings, the results showed that the online teacherpreneurs teaching experience could coincide with where they sell their curriculum resources. The data indicated that teachers with less years of experience only produced resources for TpT, and the more experienced teachers have created and share resources on additional social media educational sites. This suggests that TpT's popularity has grown over time and newer TpT teacherpreneurs do not see a need to share resources elsewhere. However, we believe this landscape may quickly change due to the roll out of Amazon Educational Insight in 2019 or e-commerce platforms run independently by individual teacher sellers, such as Shopify. We wonder if these popular online teacherpreneurs on TpT will choose to share their resources on Amazon's new platform if its popularity increases. Due to this vastly changing landscape, teacher educators must stay abreast of changes in the popularity of social media educational resources sites which is in part evidenced through online teacherpreneurs choice about where to share their developed resources.

### 5.2. Popularity

The data indicated that popularity on TpT might be influenced by factors online teacherpreneurs cannot control, like when they started on TpT, and other factors that they can control, like how many free resources they offer. This revealed that multiple elements come into play when teacher consumers view and choose to download from websites. The main influencing factor on popularity that cannot be controlled was when teachers began creating resources for TpT. The participant responses showed that the earlier a teacherpreneur started creating materials for TpT, the more downloads, likes, and followers they have. This is solely because they had more opportunities for people to view and download their materials across the existence of this website. Five participants discussed that starting when TpT was a small community was the reason for their popularity, because they were there from the start. Newer online teacherpreneurs cannot change when they begin creating and posting resources; however, these results indicate that TpT and other social media sites are built to favor longer existing

teacherpreneur profiles. Pinterest, for example, changed their search algorithm because the amount of time a pin was posted on their website influenced the top ranked items on the website. To fix this issue, Pinterest deleted the number of pins as a search element, because the older pins had a higher number of boards associated with them [30]. Therefore, this could also be true for TpT, showing us that because the materials have been on the website longer might mean that they get more views or likes.

A major influencing factor on popularity that online teacherpreneurs have control over is their product promotion. Many of the teacher participants utilized some form of social media to advertise their resources. Many used Pinterest boards, while others used Instagram and Facebook or personal blog, and yet others relied on strategic use of the features within TpT itself. Social media has become integral to many teachers' lives, and the promotion of oneself as an online teacherpreneur has become integral to their popularity. Many reported that self-promotion was necessary to get their brand out into the public, thus receiving more purchases from their TpT store. This self-promotion included writing blogs, starting Facebook groups, and having product parties. These online teacherpreneurs are curriculum resource developers, but also the promoters of their own materials.

Other than their self-promotion, another influencing factor on popularity that online teacherpreneurs can control was what they offer to their customers. For example, many online teacherpreneurs stated that "freebies" helped promote their curriculum on the website. Continually increasing their stock and making changes to their materials also brought people to their pages. However, as the results indicated, the largest majority of our teacherpreneur participants mentioned that they became popular because of the quality of their materials. Similar to Shelton and Archambault's findings, the online teacherpreneurs we surveyed are proud of their work and share it with others because they believed their developed resources represent high-quality educational materials that meet teacher needs [3].

Finally, the data indicated that a factor that could influence popularity was the strategic use of keywords when naming products. Some online teacherpreneurs recognized how the naming of their developed resources affected the teacher consumers' ability to search for and choose resources for their classroom. They believe searching for resources is influenced by the teacher's ability to type relevant keywords into a search bar and select appropriate filters (e.g., ratings, views, downloads, etc.). Thus, these online teacherpreneurs found how they name their resources influence if individuals were able to view them using the search engine showing that how products were named could influence their number of downloads.

*5.3. What Teachers Want–Discrepancy in Views*

The results indicated that the online teacherpreneurs believe that teachers want materials that align to standards but also that are free and visually appealing. They also noted that what was important to their buyers were resources that can easily be implemented in classrooms. However, this was not what we found when we surveyed 601 elementary mathematics teachers about what they looked for and valued when searching for online curriculum materials [1]. We found that teachers ranked the visual appeal of the item as one of the least important elements of the resources [1]. Schroeder, Curcio, and Lundgren also found teachers wanted connection to standards and resources to help meet their students' specific needs [22]. In addition to visual appeal, the teacher consumers also ranked price low on importance, which again shows a disconnect in what the sellers believe that the teacher consumers want and for what the teacher consumers state they are actually searching for [1]. We posit that perhaps mathematics teacher educators can help bridge this disconnect and work to communicate the need for aligning goals for creation of resources.

## 6. Limitations

This data came from a specific population that was only reported in the top 500 free elementary mathematics resources on TpT in June 2018. We are also aware that we were not able to obtain a large subset from the 321 top TpT teacherpreneurs to participate in our survey because of restrictions in place by TpT. Had we been able to obtain more responses, additional information could have been gathered

about the online teacherpreneurs demographics, as well as how they gained their popularity. We only reached out to the top TpT teacherpreneurs of free elementary mathematics resources. The individuals who are the top TpT teacherpreneurs of all K-12 resources might have other responses specifically related to the importance of using free resources to gain popularity from the consumers. It also must be noted that the data presented and discussed is limited to only the online teacherpreneurs' and teachers consumers' self-perceptions and not actual purchasing data or measurable qualities (i.e., success of their posted materials). Buying behavior could significantly be different than what teachers say they want in resources, thus TpT sellers could have a better understanding of their customers than we see in our data. Lastly, when we asked about what made you popular, the responses were subjective. Thus, the result could change based upon the population of online teacherpreneurs surveyed.

## 7. Implications

This study suggested broad implications regarding what teacherpreneurs believe constitutes quality or "good" online resources. This study also highlighted the discrepancy between what online teacherpreneurs believed their elementary teacher consumers want in their elementary mathematics products and what teacher consumers' report they actually desire. Quality is dependent on a feedback loop. One possible cycle that we see in play is first the online teacherpreneur creates elementary mathematics resources based on what they think teacher consumers want. Then they post their developed resource on a platform to be selected and downloaded, and eventually implemented by a teacher consumer in a classroom.

### 7.1. Implications for Resource Quality

Online teacherpreneurs indicated a desire to create and share quality elementary mathematics resources. In part of our larger study [24], the researchers considered the quality of these top 500 free elementary mathematics resources based on an established, well-respected framework entitled the Task Analysis Guide [31]. We found that less than 1% of these resources could be categorized as the highest level of cognitive demand while only 38% were categorized at the second highest level of demand. This illustrates the discrepancy between online teacherpreneurs' desire to create quality resources and the actual measured quality of the resources. Teacher educators must use the knowledge of this discrepancy to help support teachers who work so hard in their curriculum development. Online teacherpreneurs on TpT want to create the best possible products for their customers, and teacher educators want to support teachers in their implementation of high-level mathematical tasks that challenge students to develop their mathematical understanding. Similar to Hertel and Wessman-Enzinger's findings, our results indicate a need for teacher educators to work alongside teacherpreneurs who choose to market their resources and also to work with teacher consumers to help them learn how to choose high quality resources from social media sites such as TpT [25].

In discussing implications related to quality, it was also important to consider the perspective of the social media platform itself. From the perspective of TpT, a resource might be "good" if it generated significant traffic or comes from one of the promoted developers. It is worth noting that the TpT algorithm used to list resources after a search appears to be a black box to both the teacherpreneur and to the teacher consumer. We do not know the algorithm that the search engine on TpT uses to indicate the top items on their website. However, the results from this survey indicated that the amount of time a resource is on the website might have an impact on if it is viewed in the top 500 resources. If the algorithm does not allow for some indicator of mathematics quality, perhaps the educational research community could be influential in helping TpT to change its display algorithm. However, we are aware that motivating TpT to change their algorithm would be difficult. Since TpT is a for-profit marketplace, the consumers would have to reflect a need for this in their buying habits.

*7.2. Implications Regarding Beliefs about What Teacher Consumers Want*

One of our findings showed that online teacherpreneurs who share their resources on social media often promote their products on other social media sites or through use of TpT's promotional tools. Thus, teacher consumers may also be influenced by the amount of social media activity of a developer through advertising and visual presentation. All of these factors influence the search space and judgement of teacher consumers even when—as was the case with visual presentation—such a factor is not listed as important by the majority of teacher consumers. As part of the larger study, after we recognized the importance of visual appeal, we studied these top 500 resource picture types and found that decorative pictures that are not needed to do the mathematics were statistically significantly correlated with lower levels of cognitive demand [24]. These cute resources that online teacherpreneurs work to create because they believe this is what teacher consumers want takes away from the focus on the mathematics content and hence leads to lower demand. This may have implications for teacher training specifically in making teachers aware of how some factors are sometimes given too much weight in judging the quality of a resource.

## 8. Conclusions

Social media websites like TpT have become an influencing factor on the field of curriculum development in the United States, and the mathematics curriculum that students receive influences their understanding of the subject [1]. While this study focused specifically on elementary mathematics, we posit that the results have ramifications for all areas of K–12 education because of how much impact social media has had on teachers' lives personally and professionally. The online teacherpreneurs that are the most popular either got into the TpT business early or promoted themselves heavily through social media like Pinterest, Facebook, and Instagram. Therefore, teacher educators must learn from studies like this one to support these online mathematics teacherpreneurs in their creation of high-quality resources, because publishing companies are no longer the gatekeepers of curriculum resources and established peer reviewing processes of resources are no longer occurring. While the quality of these top resources was not indicative of high levels of cognitive demand [26], online teacherpreneurs report spending time thinking carefully about the types of resources they are creating to share with teacher consumers and they indicate a desire to create high-quality resources. We assume all stakeholders, developers, consumers, TpT, and teacher educators want teachers to have access to the best materials possible. Therefore, we hope that through studies such as these we can work together to create quality materials impacting the growing field of social media educational resources sharing sites, and thus help students better understand mathematics as a field.

**Author Contributions:** Conceptualization, A.G.S., L.K.D., and P.S.; methodology A.G.S., L.K.D., and P.S.; software, L.K.D..; validation, A.G.S., L.K.D., and P.S.; formal analysis, A.G.S., L.K.D.; investigation, A.G.S., L.K.D., and P.S.; resources, A.G.S., L.K.D., and P.S.; data curation, A.G.S., L.K.D., and P.S.; writing—original draft preparation, A.G.S.; writing—review and editing A.G.S., L.K.D., and P.S.; visualization, A.G.S., L.K.D., and P.S.; All authors have read and agreed to the published version of the manuscript.

**Funding:** This research received no external funding.

**Conflicts of Interest:** The authors declare no conflict of interest.

## Appendix A

Hello! You have been identified as someone who posts free elementary mathematics materials on Teachers Pay Teachers (TpT). As such, you are invited to participate in an anonymous research study concerning the creation of elementary mathematics materials on Teachers Pay Teachers (TpT). This research is being conducted by students, teachers, and professors at Bucknell University, James Madison University, and Somerset Public Schools. Please be assured that your responses will be kept completely confidential.

The survey should take you less than 10 mins to complete. Your participation is voluntary. You have the right to withdraw at any point during the study, for any reason, and without any

prejudice. The research team hopes to recruit approximately 500 participants. If you have any questions or concerns about this study, you may contact the Principal Investigator, Lara K. Dick, by email at lara.dick@bucknell.edu. General questions or concerns about the rights of human subjects of research may be directed to the chair of the Institutional Review Board at Bucknell: Matthew Slater at mhs016@bucknell.edu.

By clicking the button below, you acknowledge that your participation in the study is voluntary, you are at least 18 years of age, and that you are aware that you may choose to terminate your participation in the study at any time and for any reason.

- o I consent, begin survey
- o I do not consent, I do not wish to participate

Q1: Are you a current teacher?

- Yes, I currently teach elementary school.
- Yes, I currently teach something other than elementary school.
- No, I am a former teacher.
- Other: _______________

Q2: If applicable, how many years of teaching experience do you have?

**0–2**     **3–5**     **6–10**     **11–15**     **16–20**     **21–25**     **26–30**     **31–35**     **36+**

Q3: Please select the grade(s) in which you are currently teaching or have most recently taught mathematics.

Kindergarten

- 1st
- 2nd
- 3rd
- 4th
- 5th
- Other: ________

Q4: How many years of have you been producing teaching resources?

- 0
- 1
- 2
- 3
- 4
- 5
- 6
- 7
- 8
- 9
- 10
- 11
- 12
- 13
- 14
- 15

Q5: How many years of have you been producing teaching resources for Teachers Pay Teachers?

- 0
- 1
- 2
- 3
- 4
- 5
- 6
- 7
- 8
- 9
- 10

Q6: Do you create your elementary mathematics materials for Teachers Pay Teachers on your own?

- Yes
- No, I collaborate with other educators.

Q7: Have you created elementary mathematics materials for sites or companies other than Teachers Pay Teachers?

- Yes
- No

Q8: Are you affiliated with a company?

- Yes
- No

Q9: Please rank the importance of the following criteria you use when creating elementary mathematics materials that you post online.

(Drag each criteria up or down to change its rank from 1 being the most important to 9 being the least important.)

- Alignment to standards
- Fun activity
- Level of difficulty
- Perceived student engagement
- Perceived student success
- Price
- Rating
- Visual appeal
- Other

Q10: What kinds of quality checks, if any, do you implement before posting elementary mathematics materials on TpT? (e.g., mathematical, grammatical, etc.)

_______________________________________________________________________________

Q11: What resources help inspire elementary mathematics materials you create? (e.g., books, websites, curricula materials)

_______________________________________________________________________________

Q12: In our work, we found that teachers choose more materials from TpT that involve memorization and step-by-step procedures.

_________________________________________________________________________________

Q13: Does this surprise you?

- Yes
- No
- Other

Q14: Why or why not?

_________________________________________________________________________________

Q15: What makes an elementary mathematics resource "good" for TpT?

_________________________________________________________________________________

Q16: What do you think teachers look for when they are searching for elementary mathematics materials on TpT?

_________________________________________________________________________________

Q17: What do you think makes you so successful on TpT?

_________________________________________________________________________________

Q18: Please explain how your popularity and followers have grown over time. What do you believe made this occur?

_________________________________________________________________________________

## Appendix B

**Table A1.** Code Book for How Teacherpreneurs became Popular/Successful on TpT.

| Code | Definition | Example Responses from Data |
|---|---|---|
| Social Media | Participants describe using social media platforms to promote products to become popular/successful. | Having a blog and being active by posting and sharing free items as well as paid items got me attention. (2) |
| Quality | Participants describe the quality of their materials making them popular/successful. | Continuing to produce high quality materials, and if something is selling well, making other items in that same genre. (17) |
| Niche | Participants describe becoming popular/successful because they supply some form of special item needed by teachers. | At the time when I started creating products (2012), I was one of the few that differentiated activities and focused on developmentally appropriate activities such as sensory, fine motor, movement and game play. These helped me create a niche that has since been saturated by teachers that came after me, but I was one of the first … and that's where most of my followers came from when I look back. (13) |
| Early to TpT | Participants describe becoming popular/successful because they started on TpT when the platform began. | I got in on TPT early in the game. (2) |
| Large Products | Participants describe having a large number of products so people purchase them increasing their popularity/success. | My followers grow when I make new resources. (20) |
| Offer Free Stuff | Participants describe becoming popular/successful because they offer free resources on TpT. | Offering lots of freebies. (14) |

**Table A1.** *Cont.*

| Code | Definition | Example Responses from Data |
|------|-----------|----------------------------|
| Relatable | Participants describe becoming popular/successful because their product was relatable to their users. | Creating quality resources and being a current classroom teacher make me relatable. (11) |
| Don't Know | Participants describe not knowing what made them becoming popular/successful. | I'm honestly not sure. I don't follow my store too often. (1) |
| Fair Price | Participants describe becoming popular/successful because their products are offered at a good price. | Offering good value for money (14) |

## Appendix C

**Table A2.** Code Book for What Teacherpreneurs Believe Teachers Want in their Products.

| Code | Definition | Example Responses from Data |
|------|-----------|----------------------------|
| Easy to Use | Participants describe teachers wanting products that can be easily implemented in their classroom. | Ease of use, engagement, low prep. (6) |
| Visual Appeal | Participants describe teachers wanting "cute" and visually appealing resources. | Unfortunately I think teachers overvalue visual appeal on TPT. (12) |
| Price | Participants describe teachers wanting free to low price items for their classrooms. | Free resources in areas that they feel their students struggle with and for which they do not have many resources. (2) |
| Rigor | Participants describe teachers wanting activities that make their students think providing rigor. | Solid math foundation with clear instructions. Unique ways to offer practice and challenge students' thinking. (8) |
| Quality | Participants describe teachers wanting thoughtfully constructed quality materials. | Quality items that fit what they are looking for at that particular time. (3) |
| Alignment with Standards | Participants describe teachers wanting materials that align with their state's standards. | I think these are the primary reasons: *Resources that meet standards they are focusing on or will be focusing on. (4) |
| Fits a Need | Participants describe teachers wanting products that fit a specific student need. | I look for products that target that concepts my students struggle with the most. (18) |
| Engagement | Participants describe teachers wanting engaging materials that make students become hands on learners. | Something engaging for students that does not require a lot of prep and has quality content. (7) |
| Rating | Participants describe teachers wanting high rated products from TpT platform. | Rating, curriculum, I'm not sure. (23) |
| Technology | Participants describe teachers wanting resources that can be used with or on different technology. | Digital resources. Resources that use the latest technology. (21) |
| Differentiation | Participants describe teachers wanting resources that differentiate the math content for different learners. | If they are kindergarten teachers, I hope they are looking for hands-on, engaging and differentiated activities that allow their students to be successful and independent. (9) |

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
