# Peer review of "Online Mathematics Teacherpreneurs Developers on Teachers Pay Teachers: Who Are They and Why Are They Popular?"

_education, doi:10.3390/educsci10090248_

Round 1

Reviewer 1 Report

This was an interesting and well-written article addressing a timely issue in education today - teachers' use of online educational marketplaces like TpT. The framing of this work within the context of the authors' larger project on this topic makes the findings particularly powerful. Below are considerations and suggestions for revision:

Line 49 - "Shelton & Archambault found monthly earnings..." - the phrase "per year" should be removed. Their study found that "monthly earnings from TpT curriculum developers ranged from $5 to $68,000."

Line 105 - Shelton & Archambault use the term "online teacherpreneur" which is distinct from Barry et al's "teacherpreneur" - this distinction might be worth noting and specifying. Also, Shelton & Archambault do not assert that online teacherpreneur are "not affiliated with any company" as stated in line 105.

Lines 128-168: The portions of this section that focus on teachers as curators (buyers) of online curricula stands to distract from the article's focus on the creators of these resources. Another potential issue is this section's focus on curriculum implementation - earlier in the paper, the authors indicate that they are investigating TpT content creators who are not necessarily classroom teachers or "teacherpreneurs" - because of this, discussion of curriculum implementation (by teachers) within the lit review doesn't seem to be as relevant to this study. Jumping down to the implications and conclusion this mismatch becomes more pronounced, given the discussion's focus on teacher education for content creators. What about the creators who are not reached by teacher education?

Line 183-185: It is not clear if this quote accurately sums up the purpose of the study: "Thus, we are interested in learning more about TpT curriculum developers to determine if they understand and take teachers’ desires when 184 searching on social media sites for educational resources into consideration when developing their to-be-shared resources." Instead, weren't the authors looking at Who they are, why they are successful, and what they think teachers want?

Line 192 - Don't see Appendix A (the survey) at the end of this PDF.

Line 198: Should note that the rating system changed to 5 stars sometime in 2019, and that it was on 4 stars at the time of data collection.

Line 209: On June 26, 2019 all users were contacted, but when was the list populated? Because the top 500 resources are always somewhat in flux, this info would be important. Was it at the time of your previous study in 2017? Now after reading the discussion and conclusions, it seems that more context about the larger project that this study was a part of will help address this comment.

Line 217: Data analysis doesn't address how the quant data was analyzed?

Line 221 and 225: Did you mean "constant comparative method" instead of "constant comparison"?

Line 232: Appendix B and C not available at the end of this PDF (?).

Figure 1: Two of the labels are the same "Yes, I currently teach" (5.2% and 46.6%)

Results or Methods: Before presenting the results addressing the RQs, the response rate should be reported.

Lind 346: Is this surprising though? It seems like an artifact of the sampling approach. Sellers of the most downloaded free resources of all time were sampled, and these resources would be more likely to be created by sellers who were present early on, simply because they have been on the site longer to accrue more downloads. I now see you discuss this a little starting at line 444 in the discussion - some of that context would be helpful to provide in the results and/or removal of "surprising".

Line 442: Another consideration in addition to Amazon Ignite is the rising popularity of e-commerce platforms run independently by individual teacher sellers, such as shopify.

Section 444 - 459: There is an added issue that may be worth considering - did you look at the race of your respondents? A problem with TpT is that it is so white. If this status quo is perpetuated by its algorithm or even by the existence of a sorting feature "best seller," which prioritizes most downloaded resources (read: old resources), then how will racially diverse authors have much of a chance to be seen by potential buyers? What duty does TpT as a marketplace have to address this limitation?

Line 489 - what is meant by "free visually appealing resources"? Is a comma missing?

Lind 498 - A caveat that these conclusions are made with limited data based in self-perceptions (as compared to actual purchasing data) is needed. For example, teachers may say that they don't go for visual appeal, but their buying behavior may say otherwise.

Line 516 and 518-521 - I'm not sure that we can be sure that "teachers select it because it is what is available to buy" - there are likely many factors influencing their purchasing decisions, yes one of those factors is what is available, but not the only factor at play. Similarly, we do not necessarily know that "Curriculum developers judge a mathematics resource's quality as ‘good’ if they believe it is what teachers want from resources." Neither do we know that "After posting, a teacher consumer chooses what they deem to be a 519 quality resource based on the number of downloads, positive comments, or if it ranks well on the platform." These might be typical approaches or one way that users approach TpT, but we do not know that for sure based on this study and no other evidence is provided to support these statements.

Line 544 - It is worth noting that motivating TpT to change may be a huge hurdle unless TpT as a for-profit marketplace had profit to gain in making this change.

Discussion - the connections to the larger study that are brought up in the discussion are powerful. Framing this particular article as part of a larger project from the beginning and discussing the related project work in the lit review would bolster the grand conclusions from this work as a whole.

Limitation - It is also worth noting these limitations: The data was collected at one point in time - the top 500 may be (are) changing; RQ 3 asked about content creators' perceptions of content curators' perceptions. The limitations (and opportunities) of this form of non-observational measure should be noted.

Curriculum - the authors nicely clarify examples of self-made curriculum in line 40, however it may be worth explicitly noting the incomplete and/or incohesive nature of some TpT curricular materials. For many readers, a "curriculum developer" may evoke the idea of a person who writes an entire curriculum, a textbook, or a cohesive set of learning materials - not a person who designs a single worksheet that is popular on TpT.

References and Citations: Throughout there is an opportunity to connect with additional literature examining the issue of TpT, for example:

Carpenter, J. P. & Harvey, S. (2019). “There’s no referee on social media”: Challenges in educator professional social media use. Teaching and Teacher Education, 86, 1-12.

Hu, S., Torphy, K. T., Opperman, A., Jansen, K., & Lo, Y. (2018). What do teachers share within Socialized Knowledge Communities: A case of Pinterest. Journal of Professional Capital and Community, 3(2),97-122. doi: 10.1108/ JPCC-11-2017-0025

Pittard, E. A. (2017). Gettin’ a little crafty: Teachers Pay Teachers, Pinterest and neo-liberalism in new materialst feminist research. Gender and Education, 29(1), 28-47. doi: 10.1080/09540253.2016.1197380

Polikoff, M., & Dean, J. (2019). The supplemental-curriculum bazaar: Is what’s online any good? [report]. Washington, DC: Thomas B. Fordham Institute. https://fordhaminstitute.org/sites/default/files/publication/pdfs/20191210-supplemental-curriculum-bazaar0.pdf?mc_cid=782d397873&mc_eid=a11682bc98

Shelton, C. C., & Archambault, L. M. (2020). Learning from and about elite online teacherpreneurs: A qualitative examination of key characteristics, school environments, practices, and impacts. Teachers College Record, 122(7).

Schroeder et al. 2019 - cited in the manuscript but did not appear in the references

Torphy, K. & Drake, C. (2019). Educators meet the fifth estate: The role of social media in teacher training. Teachers College Record, 121(14).

Author Response

Abstract: Certain elements of the abstract are missing, including theoretical and/or conceptual perspective/s employed (other than vague "survey methodology"). It doesn't say how many individuals were surveyed or tangible results (hence, "many" isn't specific like a percentage or frequency count). Notably, it would appear (from the abtract) content accuracy or best pedagogical practices are a major consideration among those developing curriculum for TpT.This seems like a big implication.

The abstract has been completely rewritten. Thank you for the suggestions.

Introduction:In the 2019a author study, why did 90% of the elementary teachers use TpT? This seems important to discuss (in a few sentences) to substantiate needs and gaps to be addressed within this study.

Unfortunately, we did not ask the teachers in the 2018 study why they picked TpT, so we couldn't address this.

Otherwise, your claim that those TpT lesson plans are influencing curriculum is not defensible. Previous studies have demonstrated the discrepancies among the intended curriculum, enacted curriculum and achieved curriculum, meaning there is a larger game of "telephone" being played once a teacher receives a lesson plan to how it is enacted in the classroom. This framework should be provided, (see work by Glatthorn). There is no evidence suggesting that teachers follow lesson plans exactly as intended, and this is particularly true for lesson plans developed externally. Please provide a better justification. One way is to find out how those lesson plans have been used or intended to be used (e.g., for teacher professional development with how many teachers; or the lesson plans have been accessed or downloaded xx number of times).The research questions don't exactly track or map onto the need and gap illuminated, which I believe can help with the revision above.Meaning, what it does matter how they identify (per research question 1)? Does it matter that their motives are driven by altruism or profit?Further, wouldn't the 2019a author study have explored research question 2 in terms of popularity within the survey of U.S. teachers?Also, typically the research questions follow the theoretical framework such to demonstrate how theory is being fully included into the study design, analysis and interpretation.Otherwise, it may appear as a program evaluation of TpT rather than a true research study.

We added more of a discussion in the introduction explain the motives behind the study, and explain a bit more how this work is influential to the mathematics field.

Theoretical Framework:Having read the TFW, I think it provides a perspective to why teachers would seek out TpT, but doesn't fully model how or why teachers would develop curricular content for other teachers. This aspect of the theory (perhaps coupled with another to create a conceptual framework) needs to be addressed, vis-a-vis the research questions, to understand how this theory can model the real-world experiencethat is being studied here. Again, the research questions evoke ideas of motives (a.k.a. motivation), but the reader is unsure how New Literacy Theory addresses this, and further, how it contributes as a meaningful lens in the study. Meaning, how does this address issues ofidentitywhich is mentioned in research question 1? Also, math education is the call, but math education takes a large back seat to the study.Perhaps a conceptual framing is warranted to include the uniqueness of math, especially elementary mathematics, to the modeling of this specific study.

We removed NLT from the paper to help it be more streamlined with teacherpreneur research.

Literature Review...is intended to elaborate on concepts introduced within the introduction, per illuminated needs and gaps of the study.The construct "Teacherpreneur" is introduced which has not been mentioned at all until this part of the work. I agree that this is construct vital and should be meaningfully incorporated into the abstract, introduction, and theory.Further, you said that "we choose not to define the self-made curriculum developers under study as teacherpreneurs because we do not know if they are current or former teachers nor do we know if they are affiliated with a company," which is far too narrow, especially on the latter point. Any teacher, without a company, can be a teacherpreneur (per B. Barry). You also mention that TpT doesn't have any screening process, which is 1) surprising that they wouldn't have some verification measure and 2) presents a huge threat to validity for your study. It would be easier to utilize the moniker teacherpreneur, ask a demographic question on study's survey survey (as part of your selection frame), "are you now or have been recently been a full-or part-time K-12 teacher of mathematics in the U.S.?" (this is mentioned that you do this on line 189). Or, indicate that some sampled from TpT may not have been teachers (or recent teachers) as a part of the limitations section. Further, the underlying theory of teacherpreneur is part of the larger bodies of work on pathways within teacher leadership. There is a dearth of content-based teacherpreneur pieces, especially so in elementary mathematics. In total, I firmly believe that this theory (or conceptual framing of Teacherpreneurs with elementary math teachers) would be a much better way to model your study (per the research questions as they are currently written); this provides a better model for how they identify and why these individuals develop and sell content via TpT.

We decided to completely reframe the paper as you suggested to identify teacherprenurs rather than self-made curriculum developers. Thank you for the suggestion.

Methods:The response rate is low (about 12% overall), but that is to be expected based upon the nature of survey work. Prompts are in the results section, but should be identified here.Further, the prompts (in a table, preferably) should be related to the theory and research questions, such that the reader is assured that the prompts were aligned to theory and able to yield the data needed to sufficiently answer the research questions, respectively.

We made sure to include the survey in Appendix A.

Results & Discussion: I highly recommend you remove anyone who did not identify as a teacher (including the person who did not identify at all) from the study. It undermines the premise of the work (and the sampling site; TpT is the name afterall). Especially considering the users (teachers seeking lessons) have the expectation they are using content developed by teachers, not a retired military person as an example. Further, that would allow you to use the Teacherpreneur concept as your frame, which applies vital theory to the analysis and discussion, helping to situate your work within a greater contribution to the literature. There is genuine concern that this reads as a program evaluation of TpT rather than a research study.I can confidently say this because the new literacy theory is not present at all (except once as lip service on line 478), whereas your discussion actually presents a better modeling of Teacherpreneurs-based teacher leadership motives! It is highly recommended that this reframing is needed to not only bring alignment to theory and outcomes (coherency), but also provide a more meaningful contribution to literature on an important topic (teacher leadership).Further, what specific contributions to elementary math (per the call and title of the manuscript) are made here by way of this study?

We decided not to remove those individuals from the study because they identify as a teacherpreneurs because they advertise themselves as teachers creating resources for teachers. It still gives us understanding about this population of TpT teacherpreneurs. We also removed NLT from the paper to focus more on the teacherprenurs aspect.

I have some concern that its manuscript doesn't truly address the call, meaning, it is adjacent to the call since the lesson plans aren't being evaluated, rather the curriculum developers themselves. Reframing a bit to discuss the importance of them vis-a-vis the influence of curriculum (the most germane bullet point on the special call to this study) is vitally important. If shifting to more leadership (per Teacherpreneurs), would certainly eliminate this issue.

We now explicitly bring in a tie between the curriculum developers and their influence on curriculum.

I am fairly confident that the in-text citation style is numerical (not standard APA) and in order of appearance.This becomes problematic when seeing that Author 2017 is the first reference, but it does not appear in text as first (which is Author 2019). Please carefully review the author guidelines to ensure changes are made to conform to journal expectations.

We fixed all the in-text citations and references to adhere to the journal guidelines.

US should be U.S. (line 35)

It was fixed. Thank you!

Reviewer 2 Report

Dear author

The subject is well represented in the title and abstract, it provides interesting and novel information, in these moments when the virtualization of teaching has intensified implosively.

Updated and interesting review. However, it would be recommended that the object of the investigation, the TeachersPayTeachers.com portal, have a separate heading, since it is the fundamental object on which it is investigated.

The method is clearly presented and structured. It is contingent and appropriate in relation to the research questions, and offers the academic reader a sufficient degree of transparency.

The results, although exclusively descriptive in the quantitative part, provide data of interest. In the qualitative part, it would be appropriate to reduce the quotes of the actors.

Discussion, implications and conclusion are well structured and interesting.

References require an important modification, because in addition to not following the journal's norm, there is an inconsistency between the text citation system and the ordering system in the references section. They should also review some small formal aspects, such as the use of the final point that the author / s place in the sentence before the quote and not after as it should.

Regards

Author Response

Updated and interesting review. However, it would be recommended that the object of the investigation, the TeachersPayTeachers.com portal, have a separate heading, since it is the fundamental object on which it is investigated.

We made the section describing the platform as a separate heading to identify the importance of the concept.

The results, although exclusively descriptive in the quantitative part, provide data of interest. In the qualitative part, it would be appropriate to reduce the quotes of the actors.

We decreased the length of many of the quotes.

References require an important modification, because in addition to not following the journal's norm, there is an inconsistency between the text citation system and the ordering system in the references section. They should also review some small formal aspects, such as the use of the final point that the author / s place in the sentence before the quote and not after as it should.

All references and citations are now completed in the journal's style requirements.

Reviewer 3 Report

Please use the following information to strengthen your manuscript:

Abstract: Certain elements of the abstract are missing, including theoretical and/or conceptual perspective/s employed (other than vague "survey methodology"). It doesn't say how many individuals were surveyed or tangible results (hence, "many" isn't specific like a percentage or frequency count). Notably, it would appear (from the abtract) content accuracy or best pedagogical practices are a major consideration among those developing curriculum for TpT.  This seems like a big implication.

Introduction:  In the 2019a author study, why did 90% of the elementary teachers use TpT? This seems important to discuss (in a few sentences) to substantiate needs and gaps to be addressed within this study. 

Otherwise, your claim that those TpT lesson plans are influencing curriculum is not defensible. Previous studies have demonstrated the discrepancies among the intended curriculum, enacted curriculum and achieved curriculum, meaning there is a larger game of "telephone" being played once a teacher receives a lesson plan to how it is enacted in the classroom. This framework should be provided, (see work by Glatthorn). There is no evidence suggesting that teachers follow lesson plans exactly as intended, and this is particularly true for lesson plans developed externally. Please provide a better justification. One way is to find out how those lesson plans have been used or intended to be used (e.g., for teacher professional development with how many teachers; or the lesson plans have been accessed or downloaded xx number of times).

The research questions don't exactly track or map onto the need and gap illuminated, which I believe can help with the revision above.  Meaning, what it does matter how they identify (per research question 1)? Does it matter that their motives are driven by altruism or profit?  Further, wouldn't the 2019a author study have explored research question 2 in terms of popularity within the survey of U.S. teachers?  Also, typically the research questions follow the theoretical framework such to demonstrate how theory is being fully included into the study design, analysis and interpretation.  Otherwise, it may appear as a program evaluation of TpT rather than a true research study. 

Theoretical Framework:  Having read the TFW, I think it provides a perspective to why teachers would seek out TpT, but doesn't fully model how or why teachers would develop curricular content for other teachers. This aspect of the theory (perhaps coupled with another to create a conceptual framework) needs to be addressed, vis-a-vis the research questions, to understand how this theory can model the real-world experience that is being studied here. Again, the research questions evoke ideas of motives (a.k.a. motivation), but the reader is unsure how New Literacy Theory addresses this, and further, how it contributes as a meaningful lens in the study. Meaning, how does this address issues of identity which is mentioned in research question 1? Also, math education is the call, but math education takes a large back seat to the study.  Perhaps a conceptual framing is warranted to include the uniqueness of math, especially elementary mathematics, to the modeling of this specific study.

Literature Review...is intended to elaborate on concepts introduced within the introduction, per illuminated needs and gaps of the study.  The construct "Teacherpreneur" is introduced which has not been mentioned at all until this part of the work. I agree that this is construct vital and should be meaningfully incorporated into the abstract, introduction, and theory.  Further, you said that "we choose not to define the self-made curriculum developers under study as teacherpreneurs because we do not know if they are current or former teachers nor do we know if they are affiliated with a company," which is far too narrow, especially on the latter point. Any teacher, without a company, can be a teacherpreneur (per B. Barry). You also mention that TpT doesn't have any screening process, which is 1) surprising that they wouldn't have some verification measure and 2) presents a huge threat to validity for your study. It would be easier to utilize the moniker teacherpreneur, ask a demographic question on study's survey survey (as part of your selection frame), "are you now or have been recently been a full- or part-time K-12 teacher of mathematics in the U.S.?" (this is mentioned that you do this on line 189). Or, indicate that some sampled from TpT may not have been teachers (or recent teachers) as a part of the limitations section. Further, the underlying theory of teacherpreneur is part of the larger bodies of work on pathways within teacher leadership. There is a dearth of content-based teacherpreneur pieces, especially so in elementary mathematics. In total, I firmly believe that this theory (or conceptual framing of Teacherpreneurs with elementary math teachers) would be a much better way to model your study (per the research questions as they are currently written); this provides a better model for how they identify and why these individuals develop and sell content via TpT.  

Methods:  The response rate is low (about 12% overall), but that is to be expected based upon the nature of survey work. Prompts are in the results section, but should be identified here.  Further, the prompts (in a table, preferably) should be related to the theory and research questions, such that the reader is assured that the prompts were aligned to theory and able to yield the data needed to sufficiently answer the research questions, respectively. 

Results & Discussion: I highly recommend you remove anyone who did not identify as a teacher (including the person who did not identify at all) from the study. It undermines the premise of the work (and the sampling site; TpT is the name afterall). Especially considering the users (teachers seeking lessons) have the expectation they are using content developed by teachers, not a retired military person as an example. Further, that would allow you to use the Teacherpreneur concept as your frame, which applies vital theory to the analysis and discussion, helping to situate your work within a greater contribution to the literature. There is genuine concern that this reads as a program evaluation of TpT rather than a research study.  I can confidently say this because the new literacy theory is not present at all (except once as lip service on line 478), whereas your discussion actually presents a better modeling of Teacherpreneurs-based teacher leadership motives! It is highly recommended that this reframing is needed to not only bring alignment to theory and outcomes (coherency), but also provide a more meaningful contribution to literature on an important topic (teacher leadership).  Further, what specific contributions to elementary math (per the call and title of the manuscript) are made here by way of this study?

Other Issues:

I have some concern that it manuscript doesn't truly address the call, meaning, it is adjacent to the call since the lesson plans aren't being evaluated, rather the curriculum developers themselves. Reframing a bit to discuss the importance of them vis-a-vis the influence of curriculum (the most germane bullet point on the special call to this study) is vitally important. If shifting to more leadership (per Teacherpreneurs), would certainly eliminate this issue.

I am fairly confident that the in-text citation style is numerical (not standard APA) and in order of appearance.  This becomes problematic when seeing that Author 2017 is the first reference, but it does not appear in text as first (which is Author 2019).  Please carefully review the author guidelines to ensure changes are made to conform to journal expectations.

US should be U.S. (line 35)

Author Response

Line 49 -"Shelton & Archambault found monthly earnings..." -the phrase "per year" should be removed. Their study found that "monthly earnings from TpT curriculum developers ranged from $5 to $68,000."

Thank you for catching this. We removed the language.

Line 105 -Shelton & Archambault use the term "online teacherpreneur" which is distinct from Barry et al's "teacherpreneur" -this distinction might be worth noting and specifying. Also, Shelton & Archambault do not assert that online teacherpreneur are "not affiliated with any company" as stated in line 105.

We changed references of teacherpreneurs to online teacherpreneurs. We also described the differences between the terms in a new literature review section called online teacherpreneurs. We also removed the language saying that online teacherpreneurs were not affiliated with companies.

Lines 128-168: The portions of this section that focus on teachers as curators (buyers) of online curricula stands to distract from the article's focus on the creators of these resources. Another potential issue is this section's focus on curriculum implementation -earlier in the paper, the authors indicate that they are investigating TpT content creators who are not necessarily classroom teachers or "teacherpreneurs" -because of this, discussion of curriculum implementation (by teachers) within the lit review doesn't seem to be as relevant to this study. Jumping down to the implications and conclusion this mismatch becomes more pronounced, given the discussion's focus on teacher education for content creators. What about the creators who are not reached by teacher education?

Very good point. We removed the section about the buyers of resources to focus more on the online teacherpreneurs. We also made sure to state that we did not know about non-teacher developers and highlighted what we did know about teacherpreneurs.

Line 183-185: It is not clear if this quote accurately sums up the purpose of the study: "Thus, we are interested in learning more about TpT curriculum developers to determine if they understand and take teachers’ desires when 184 searchingon social media sites for educational resources into consideration when developing their to-be-shared resources." Instead, weren't the authors looking at Who they are, why they are successful, and what they think teachers want?

Thank you for catching this. We changed the language to articulate the correct purpose of the study.

Line 192 -Don't see Appendix A (the survey) at the end of this PDF.

We added Appendix A.

Line 198: Should note that the rating system changed to 5 stars sometime in 2019, and that it was on 4 stars at the time of data collection.

We made note of this recent change in the methods section.

Line 209: On June 26, 2019 all users were contacted, but when was the list populated? Because the top 500 resources are always somewhat influx, this info would be important. Was it at the time of your previous study in 2017? Now after reading the discussion and conclusions, it seems that more context about the larger project that this study was a part of will help address this comment.

We collected the data in June 6th, 2018, and now describe this in more detail in the sampling techniques section of the paper.

Line 217: Data analysis doesn't address how the quant data was analyzed?

We used descriptive statistics which are now described more fully in the data analysis section.

Line 221 and 225: Did you mean "constant comparative method" instead of "constant comparison"?

Thank you for catching our error. Yes, it was the constant comparative method.

Line 232: Appendix B and C not available at the end of this PDF (?).

Appendix B and C have been included in the revisions.

Figure 1: Two of the labels are the same "Yes, I currently teach" (5.2% and 46.6%)

The language was fixed in the graphic.

Results or Methods: Before presenting the results addressing the RQs, the response rate should be reported.

The response rate is now reported for each survey question.

Lind 346: Is this surprising though? It seems like an artifact of the sampling approach. Sellers of the most downloaded free resources of all time were sampled, and these resources would be more likely to be created by sellers who were present early on, simply because they have been on the site longer to accrue more downloads. I now see you discuss this a little starting at line 444 in the discussion -some of that context would be helpful to provide in the results and/or removal of "surprising".

We removed the word "surprising".

Line 442: Another consideration in addition to Amazon Ignite is the rising popularity of e-commerce platforms run independently by individual teacher sellers, such as shopify.

Thank you for the suggestion. We added it to the discussion.

Section 444 -459: There is an added issue that may be worth considering -did you look at the race of your respondents? A problem with TpT is that it is so white. If this status quo is perpetuated by its algorithm or even by the existence of a sorting feature "best seller," which prioritizes most downloaded resources (read: old resources), then how will racially diverse authors have much of a chance to be seen by potential buyers? What duty does TpT as a marketplace have to address this limitation?

We agree that this is an issue worth study, but we did not ask the survey participants to identify age, race, or gender, so we are not able to address this in this investigation. If you feel like it should be addressed, we could add in a discussion of this in the limitations section.

Line 489 -what is meant by "free visually appealing resources"? Is a comma missing?Yes, these were two separate ideas.

Thank you for catching that point of confusion.

Lind 498 -A caveat that these conclusions are made with limited data based in self-perceptions (as compared to actual purchasing data) is needed. For example, teachers may say that they don't go for visual appeal, but their buying behavior may say otherwise.

Good point. We addressed this in the limitation section.

Line 516 and 518-521 -I'm not sure that we can be sure that "teachers select it because it is what is available to buy" -there are likely many factors influencing their purchasing decisions, yes one of those factors is what is available, but not the only factor at play. Similarly, we do not necessarily know that "Curriculum developers judge a mathematics resource's quality as ‘good’ if they believe it is what teachers want from resources." Neither do we know that "After posting, a teacher consumer chooses what they deem to be a 519 quality resource based on the number of downloads, positive comments, or if it ranks well on the platform." These might be typical approaches or one way that users approach TpT, but we do not know that for sure based on this study and no other evidence is provided to support these statements.

We more clearly address this in the introduction of the paper and removed the discussion regarding potential purchasing behaviors from the discussion as it distracted from the main points of the study and discussion.

Line 544 -It is worth noting that motivating TpT to change may be a huge hurdle unless TpT as a for-profit marketplace had profit to gain in making this change.

We added it to the discussion on profit gain as motivation of changing the algorithm.

Discussion -the connections to the larger study that are brought up in the discussion are powerful. Framing this particular article as part of a larger project from the beginning and discussing the related project work in the lit review would bolster the grand conclusions from this work as a whole.

We now include a much more detailed description of our research agenda in a separate section in the literature review explaining the importance of this research in describing teachers' online resources use habits.

Limitation -It is also worth noting these limitations: The data was collected at one point in time -the top 500 may be (are) changing; RQ 3 asked about content creators' perceptions of content curators' perceptions.The limitations (and opportunities) of this form of non-observational measure should be noted.

We were able to describe these limitations in the paper. Thank you for your suggestions.

Curriculum -the authors nicely clarify examples of self-made curriculum in line 40, however it may be worth explicitly noting the incomplete and/or in cohesive nature of some TpT curricular materials. For many readers, a "curriculum developer" may evoke the idea of a person who writes an entire curriculum, a textbook, or a cohesive set of learning materials -not a person who designs a single worksheet that is popular on TpT.

We have included a discussion of this in the curriculum development and classroom implementation section of the literature review.

Round 2

Reviewer 3 Report

I have reviewed the comments to authors and revised manuscript remitted.  The authors have addressed mine and the other reviewers comments.  There may be additional formatting issues (i.e., are block quotations are single spaced (rather than double) per the journal's guidelines?), but it is an accept for me.